# Exploring the relationship between motor visual proficiency and performance metrics in elite skeet shooters: An in-depth analysis

**Dongxu Gao[1,2], Beishi Hu[1], Tinggang Yuan[1], Qingshou Guo[1], Pengfei Wei[1], Yang Wu[3]***, **Chao Chen[1]**

**1** Sports Training Science Laboratory, China Institute of Sport Science, Beijing, China, **2** School of Athletic Performance, Shanghai University of Sport, Shanghai, China, **3** Sport Recreation and Tourism College, Beijing Sport University, Beijing, China

* bigtree8895@163.com

## Abstract

### Background

Motor vision ability entails using eyesight to collect and interpret information, such as tracking moving targets, understanding spatial relationships, predicting object movement, making decisions, and taking actions. This study aimed to explore the relationship between the visual skills of elite skeet shooters and their competition performance.

### Methods

In this cross-sectional study (n = 42), elite skeet shooters from the Chinese National Clay Target Shooting Training Team with a mean age of 25.63 ± 6.2 years and an average training years of 7.64 ± 3.43 participated. The fundamental visual ability variables were measured using the Senaptec system, and their specialized visual ability indices were measured during target viewing tasks using aSee Glasses. The relationship between visual acuity test indices and sports performance was analyzed using correlation coefficients and multiple linear regression analysis.

### Results

The strongest positive association was observed between Perceived Range (PS) and sports performance (r = 0.486, p < 0.001), indicating that athletes with a higher perceptual range tend to perform better. Moderate positive correlations (r = 0.333 to r = 0.362, p < 0.001) were also found for Visual Clarity (VCR), Near/Far Switching (NFQ-SCORE), Multi-target Tracking Speed (MOTSPEED), and Go/No-Go Score (GNG-SCORE), suggesting these visual skills are beneficial for performance. Conversely, a strong negative correlation was noted between Near/Far Switching Reaction Time

**Data availability statement:** All relevant data are in the manuscript and its supporting information file.

**Funding:** This work is supported by China Institute of Sport Science Basic Scientific Research Business Funding Project,(Basic 23-39).

**Competing interests:** The authors have declared that no competing interests exist.

(NFQFRT) and performance (r = −0.510, p < 0.001), highlighting that slower reaction times are detrimental. Additionally, Target Capture (TC), Depth Perception (DPP), and Eye-Hand Coordination (EHC_RT) showed moderate negative correlations (r = −0.425 to r = −0.241, p < 0.001) with performance. The regression model explained 76.7% of the variance in athletes' specialized performance (R² = 0.752, F = 49.692, p < 0.001), with key predictors including NFQFRT, EHC_RT, PS, and several specialized visual skills.

## Conclusion

The visual abilities of elite skeet shooters significantly affect their performance, underscoring the importance of perceptual range, reaction time, and specialized visual skills.

---

## Introduction

Motor vision ability pertains to the capacity of individuals to utilize their visual system for the reception, comprehension, and processing of information while engaged in sporting activities. This encompasses various facets, including the perception of moving targets, spatial orientation, and prediction of movement trajectories, decision-making judgments, and execution [1]. Clay target shooting imposes substantial demands on athletes' motor visual ability, necessitating that the speed of visual information processing at the moment of striking approaches the limits of the vestibular-ocular reflex system for elite skeet athletes. These athletes are afforded only a few milliseconds to assess and make decisions regarding the clay target and other crucial visual information [2]. Studies have revealed that high-level athletes' exhibit enhanced abilities in prediction, pattern recognition, and visual search compared to average athletes [3,4] Elite skeet shooters also possess faster reaction times, as well as enhanced visual acquisition and target tracking capabilities [5,6]. However, further exploration is required to ascertain precisely which motor visual abilities are more crucial in supporting motor performance.

Previous research has examined the visual response to intercepting movements of targets similar to those in skeet shooting [7], dynamic stereoscopic acuity[8], hallucinogens on the resting brain [9], visual clay target discrimination and sporting performance [10]. Correlation analyses have been conducted, but discrepancies in results arise due to different assessment instruments and criteria used in these studies [11,12]. The Senaptec system, formerly known as the Nike SPARQ Sensory Training Station, is capable of performing a full range of motor vision tests and has been tested for reliability [12,13]. The system has also been used to some extent in the study of target interceptability projects in baseball [14,15], ice hockey [16] and softball [17]. The skeet shooting program also had a study that used the assessment system to conduct a correlation analysis between basic visual ability and specialty, demonstrating a higher correlation between visual information processing ability and specialty performance [16,18]. While the study elucidates the relationship between

various basic visual abilities and specialization, it is not without limitations owing to a relative lack of ecological validity stemming from discrepancies between the stimulus materials employed and the specialized scenarios encountered in actual competition. Presently, there is an absence of a comprehensive evaluation study examining the progression from basic to specific motor visual abilities in clay target shooting.

Most evaluations of specialized visual abilities in skeet shooting use experimental eye-movement studies targeting visual search patterns [6,19]. Visual search in motion refers to the process by which subjects rapidly search for and select the information they need and process it to form useful information in complex motor situations [20,21]. Eye tracking technology is also widely used in the sports field to study visual search patterns in different projects [22]. By utilizing eye movement equipment, it is possible to record the athlete's gaze point, gaze duration, gaze shifts, and other eye movement indices within the search area during the movement process. This effectively reflects the athlete's cognitive regulatory activities during the visual search process. Previous research in this domain has primarily adopted the expert-novice paradigm to compare differences in information recognition and processing between sports experts and novices. The majority of these studies have demonstrated that skilled athletes exhibit faster visual information processing, higher prediction accuracy, and enhanced responsiveness [21,22]. These abilities are manifested through more efficient visual searches and more rational search strategies, as evidenced by oculomotor characteristics [23]. Previous studies have also found that skeet shooters have a shorter latency to eye jumps compared to others [24], and gaze stability is greater under interference conditions [18]. However, the expert-novice comparison paradigm is relatively simplistic, which results in certain limitations. For example, it merely highlights some of the strengths exhibited by experts without delving into the underlying mechanisms or providing a comprehensive understanding of the differences between experts and novices [24,18]. In this study, the aim is to address these limitations by adopting a more in-depth approach to analyze the visual search patterns and their correlation with specialized performance. By doing so, it is hoped to provide a more detailed and nuanced understanding of the factors that contribute to high-level performance in skeet shooting, thereby adding to the novelty of the study.

Few studies have explored the correlation between visual search pattern metrics and specialized performance. A considerable number of these studies have solely focused on particular aspects of the visual search pattern or utilized limited sample sizes, leading to incomplete or inconclusive results. Moreover, they frequently fail to conduct a comprehensive analysis of the underlying mechanisms linking visual search patterns to specialized performance, resulting in a lack of understanding of the intrinsic relationship between these metrics and specialized sports performance. Consequently, there is a dearth of effective data to support the development of key visual abilities crucial for specialized sports.

The skeet shooting technique emphasizes target reading, and the proficiency of this target reading technique directly influences the speed of initiating and transporting the gun, as well as the likelihood of hitting the clay target [25]. This process, wherein athletes must rely on their vision to rapidly search for valuable information in order to anticipate and promptly react to potential situations on the field, constitutes an effective stimulus material for assessing the specialized visual abilities of skeet shooters. By exploring and comprehensively evaluating the visual search pattern characteristics of elite skeet shooters across various scenarios, and analyzing the correlation between visual indices and specialized performance, this study aims to gain a deeper understanding of the role of visual search ability in different contexts on athletes' specialized performance. Furthermore, it seeks to cognize the visual search characteristics of high-level skeet shooters, thereby providing a reliable foundation for the targeted development of specialized visual abilities and enhancing athletes' competitive performance. Ultimately, this research endeavors to offer a valuable basis and useful reference for improving the overall performance of skeet shooters through targeted visual ability training.

This paper explores the relationship between visual ability indices of elite skeet shooting athletes from the China National Clay Target Shooting Training Team and their sports performance to provide guidance for coaches and athletes in visual evaluation, training, and future selection. By elucidating these relationships, the study aims to contribute to the

development of more effective training programs that enhance visual skills critical for success in skeet shooting, thereby supporting athletes in achieving higher levels of performance.

## Materials and methods

### Study setting and ethical approval

Ethical approval for the study was obtained from the Chinese Institute of Sport Science Ethics Committee (CISSEC), reference number 2023.09.02, dated September 2, 2023. The study was conducted in accordance with the Declaration of Helsinki. In addition, written informed consent was obtained from all participants who take part in the study. The research was initially carried out at the sports vision laboratory of the Chinese National Clay Target Shooting Training Team camp from November 20, 2023 to April 20, 2024.

### Study design and participants

A cross-sectional study design was used to explore the relationship between the visual skills of elite skeet shooters and their competition performance of (n = 42; 20 males, 22 females), elite skeet shooters from the Chinese National Clay Target Shooting Training Team with a mean age of 25.63 ± 6.2 years and an average training years of 7.64 ± 3.43. Subjects were selected based on their team membership and participation in the National Shooting Championships or higher-level events. Eligible athletes had to be teammates involved in these competitions, with no specific exclusion criteria defined. The subjects trained under the National Clay Target Shooting Collective Training Team's regimen, practicing at least five days a week for a minimum of three hours daily. Performance was assessed using the Senaptec system for fundamental visual abilities and aSee Glasses for specialized visual tasks. Mean scores from three qualification rounds of the World Cup selection matches were used to evaluate athletic performance.

The decision on the number of competitions was thoroughly justified through research and endorsed by coaches. The participants selected for this study were skeet shooters who were required to complete a total of five rounds of shooting in the qualifying rounds, with each round consisting of 125 targets. The results of the qualifying rounds were deemed the most accurate method for assessing the competitive level of the skeet shooters, irrespective of whether the evaluation was based on the stability of their technical execution or their overall athletic prowess. Therefore, the study counted the subjects' performance in the three qualifying races in Cyprus, Putian and Qingliu stations of the Egyptian World Cup Selection after the motor visual ability test, and calculated the mean value as the final indicator of the specialized performance.

### Procedures

All tests, including the basic visual ability test and the specialized visual ability test, were conducted at the sports vision laboratory of the National Clay Target Shooting Team. To prevent conflicts with the competition schedule and mitigate the effects of fatigue from high-intensity competitions on test results, the assessments were carried out three weeks prior to the selection match for the first World Cup. A minimum interval of three days between the basic visual ability test and the specialized visual ability test was maintained to reduce the likelihood of test interactions affecting outcomes.

Athletes were informed to avoid alcohol consumption, late nights, and the use of electronic devices for one hour before the tests. Additionally, all participants were screened to ensure they had no history of epilepsy, stroke, migraine, or other related conditions. The motor vision ability tests were completed within ten days. During this period, the basic visual ability test was conducted using the Senaptec system, while the specialized visual ability test utilized the aSee Glasses (7INVENSUN, Beijing, China), a spectacle-type eye-tracking device with a resolution of 0.50° and an accuracy of 0.5°–1.0°. The system's high-frequency sampling rate (120 Hz) and precision make it suitable for capturing dynamic gaze behaviors in sport-specific scenarios. Since, the Senaptec system and aSee Glasses complement one another: Senaptec

quantifies foundational visual skills (e.g., static acuity, contrast sensitivity), while aSee Glasses capture sport-specific visual behaviors (e.g., gaze stability during target tracking). This dual approach bridges the gap between laboratory-based assessments and real-world performance, ensuring ecological validity. The basic visual ability test included ten items, which are detailed in the order in which the test system was set up (S1 Appendix). The testing time varies depending on the number of participants and the nature of the test. Participants were allowed to rest prior to conducting these tests if needed. Data obtained using the Senaptec system for visual acuity and contrast sensitivity tests, conducted at the National Clay Target Shooting Team laboratory. Adapted from Senaptec manual [26] and prepared by the authors.

For the visual clarity test, athletes position themselves closer to a touchscreen display on a mobile device to interact with the interface. The athlete's task is to identify a central dot on the screen. This test aims to assess the athlete's static auditory/visual (A/V) perception by determining the direction of the dot and their response to it. This test evaluates the athlete's ability to perceive clear details at a fixed distance, which is crucial for understanding their visual clarity and response time (Fig 1).

In the hand-eye coordination test, the athlete interacts with targets that appear on a large screen. The objective is to touch circles as they change color. This test evaluates the speed and accuracy of the athlete's responses to the color changes while assessing their visual guidance and motor coordination (Fig 2).

In the special vision ability test using the aSee Glasses eye-tracking device, the athlete wears glasses designed to monitor eye movements in response to dynamic stimuli. This setup evaluates the athlete's capacity to process visual information while tracking targets, ultimately revealing real visual search patterns and reaction times scenarios (Fig 3).

The experimental video materials were recorded at Range A of the National Flying Clay Target Shooting training site in Putian using a dedicated SONY sports camera. The shooting parameters included manual and HD modes, set as follows: a shutter speed of 1/1,000, a resolution of 4K, a frame rate of 50 fps, and an automatic white balance. The camera was positioned at a height of 1.6 meters, centered on the fourth target location, to capture video stimulus material. This placement aimed to ensure a consistent perspective and presentation of images that matched the athletes' visual experience of the target (Fig 4). The video recording took place between 3 PM and 5 PM, during which the lighting conditions were relatively soft. Additionally, recordings were conducted under weather conditions where wind speeds were below Level

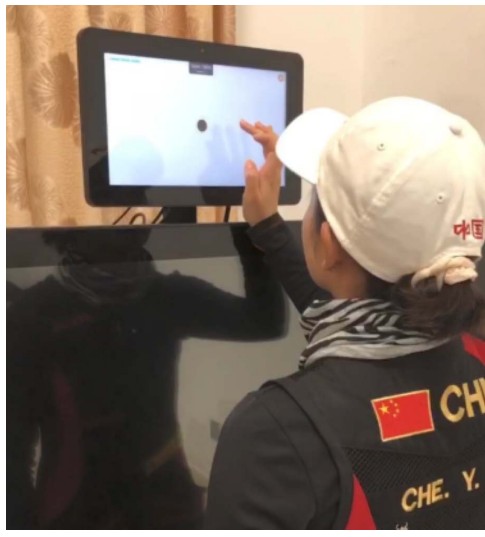

**Fig 1. Perceived Range (PS) test.**

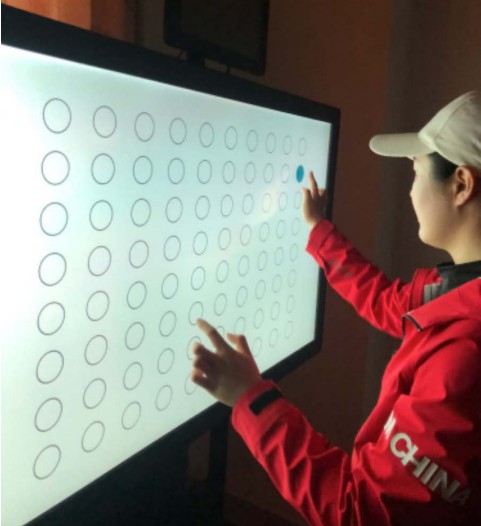

**Fig 2. Eye-Hand Coordination (EHC) test.**

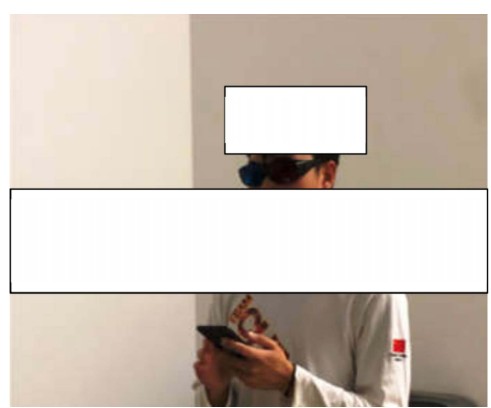

**Fig 3. Depth Perception (DP) test.**

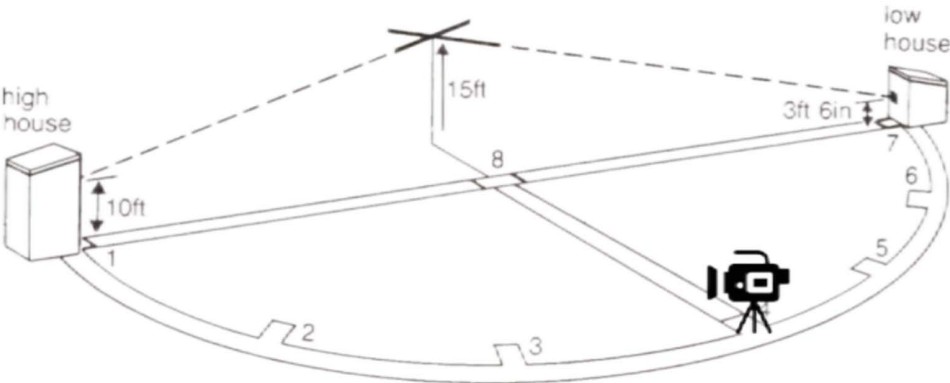

**Fig 4. Test material acquisition diagram.**

3 to maintain the stability of the clay target's flight trajectory, thereby optimizing the quality of the collected material. The camera was positioned at station 4 to capture the target's flight material.

We recorded 20 videos showcasing various flight patterns, including high and low platform single and double targets. National coaches, referees, and experts evaluated these videos to select 15 suitable participants. The experiments used the aSee Glasses eye-tracking device, which captures eye movements at a 120 Hz sampling rate and accurately records data. Participants watched the videos on a large 55-inch screen 1.5 meters away to simulate their viewing angle during competitions. The eye-tracking device was carefully calibrated to ensure precise data collection (Fig 5).

Before the main experiment, participants underwent a pre-test with two video sets to familiarize themselves with the setup. During the experiment, they watched 15 randomly selected videos (out of 60) and pressed the space bar when they saw the clay target, measuring their reaction speed. Each video followed a short break to rest and prepare for the next one. The aSee Studio software recorded the reaction times, capturing how quickly participants responded to the targets. This setup provided valuable insights into their visual processing and reaction capabilities. This version maintains the essential details while making the text more engaging and easier to read.

## Data analysis

Descriptive statistics were expressed either as mean±standard deviation (Mean±SD). These scores, reflecting the athletes' real competitive levels, were used as the dependent variable in correlation and regression analyses to determine the relationship between motor visual ability and specialized performance. The mean number of targets hit in all three rounds quantified the athletes' performance, with higher scores indicating better performance. The relationship between visual acuity test indices and sports performance was analyzed using correlation coefficients and multiple linear regression analysis. All statistical analyses were performed using IBM-SPSS version 26 (IBM, Armonk, NY, United States of America). All reported p-values are two tailed and confidence intervals are calculated at 5% alpha value.

## Results

The study involved n=42 (20 males, 22 females) elite skeet shooters from the Chinese National Clay Target Shooting Training Team with a mean age of 25.63±6.2 years and an average training years of 7.64±3.43. The average score of the specialized results in the three World Cup selection races was 112.37±4.69, with a 95% confidence interval of 111.73 to 113.00. Visual clarity (VC) scores were high, with the right eye (VCR) at 4.83±0.23, left eye (VCL) at 4.87±0.08, and binocular (VCB) at 4.96±0.04 on a 5-point scale. Contrast sensitivity (CS) was 2.00±0.02 for CS6/logCS and 1.98±0.12 for CS18/logCS. Depth perception (DP) thresholds were 106.08±14.13 arcsec for binocular (DPP), 131.55±10.98 arcsec

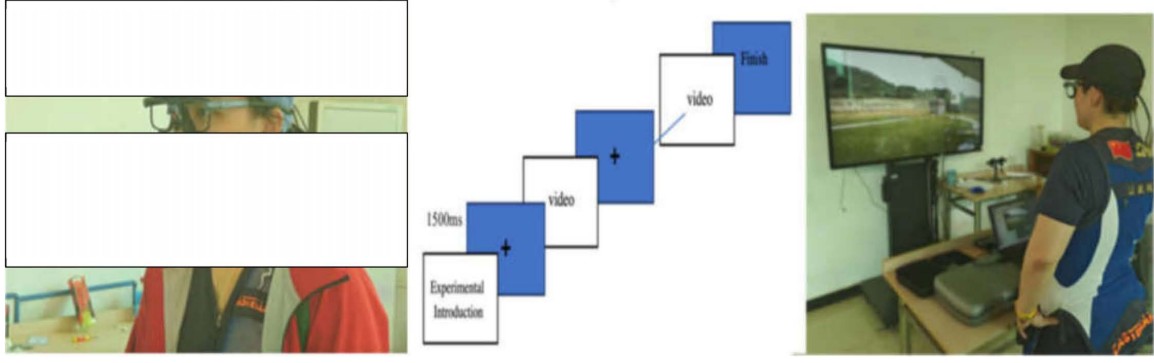

**Fig 5. Testing Experiment Scene Diagram.**

**Table 1. Correlations between basic motor visual ability test indicators and specialized performance.**

| Predictive Variables | r | Y Specific Achievements | | | | |
|---|---|---|---|---|---|---|
| | | R² | Calibrate R² | Durbin Watson | F | t |
| VC_R (5-point scale) the closer to 5.0 the better | .333** | 0.111 | 0.106 | 1.907 | 25.887*** | 5.088*** |
| VC_L (5-point scale) the closer to 5.0 the better | 0.051 | 0.003 | −0.002 | 2.033 | 0.546 | 0.739 |
| VC_B (5-point scale) the closer to 5.0 the better | 0.046 | 0.002 | −0.003 | 2.022 | 0.448 | 0.669 |
| CS_6/logCS | 0.015 | 0.000 | −0.005 | 2.029 | 0.049 | 0.222 |
| CS_18/logCS | 0.123 | 0.015 | 0.010 | 2.063 | 3.175 | 1.782 |
| DP_P/arcsec | −.377** | 0.142 | 0.138 | 2.043 | 34.457*** | −5.870*** |
| DP_L/arcsec | −0.049 | 0.002 | −0.002 | 2.033 | 0.492 | −0.701 |
| DP_R/arcsec | −0.061 | 0.004 | −0.001 | 2.021 | 0.774 | −0.880 |
| NFQ_SCORE | .360** | 0.130 | 0.125 | 1.860 | 30.947*** | 5.563*** |
| NFQ_N_RT/ms | −0.056 | 0.003 | −0.002 | 2.027 | 0.659 | −0.812 |
| NFQ_F_RT/ms | −.510** | 0.260 | 0.256 | 1.845 | 72.938*** | −8.540*** |
| TC/ms | −.403** | 0.162 | 0.158 | 2.086 | 40.342*** | −6.352*** |
| PS/Piece | .486** | 0.237 | 0.233 | 1.964 | 64.466*** | 8.029*** |
| MOT_P_S | 0.121 | 0.015 | 0.010 | 2.055 | 3.102 | 1.761 |
| MOT_C_S | 0.011 | 0.000 | −0.005 | 2.028 | 0.026 | 0.162 |
| MOT_OBJ/Piece | .273** | 0.075 | 0.070 | 1.988 | 16.762*** | 4.094*** |
| MOT_SPEED/(Degrees/s) | .351** | 0.123 | 0.119 | 1.986 | 29.201*** | 5.404*** |
| EHC_T/ms | −0.093 | 0.009 | 0.004 | 2.012 | 1.818 | −1.348 |
| EHC_RT/ms | −.425** | 0.180 | 0.177 | 2.010 | 45.809*** | −6.768*** |
| EHC_C_RT/ms | −.412** | 0.170 | 0.166 | 2.135 | 42.583*** | −6.526*** |
| EHC_P_RT/ms | −.316** | 0.100 | 0.095 | 2.138 | 23.057*** | −4.802*** |
| GNG__SCORE | .362** | 0.131 | 0.127 | 2.061 | 31.311*** | 5.596*** |
| GNG_G_HIT/Piece | .362** | 0.131 | 0.127 | 2.047 | 31.370*** | 5.601*** |
| GNG_R_HIT/Piece | −.241** | 0.046 | 0.041 | 2.009 | 9.969** | −3.157** |
| RT_A/ms | −0.042 | 0.002 | −0.003 | 2.037 | 0.370 | −0.608 |
| RT__D/ms | −.478** | 0.228 | 0.224 | 2.010 | 61.475*** | −7.841*** |
| RT_ND/ms | −0.108 | 0.012 | 0.007 | 2.038 | 2.457 | −1.568 |

Note: ***p<0.001**P<0.01*P<0.05 (below). VC_R=Visual Clarity (Right eye); VC_L=Visual Clarity (Left eye); VC_B=Visual Clarity (Binocular); CS=Contrast Sensitivity; DP=Depth of Perception; NFQ = Near/Far Switching; PS=Perceived Range; MOT=Multi-target Tracking; RT=Reaction Time; EHC=Eye-Hand Coordination; GNG=Go/No-Go Score.

for the left eye (DPL), and 126.34±15.25 arcsec for the right eye (DPR). The details of each outcome indicator's descriptive information are explained in supplementary file (S2 Table and S3 Table).

## Association between motor visual ability and performance

Table 1 shows the correlations between basic motor visual ability test indicators and specialized performance. The strongest positive correlation was found between Perceived Range (PS) and sports performance (r=0.486, p<0.001), indicating that athletes with a higher perceptual range tend to perform better. Moderate positive correlations were observed for Visual Clarity (VCR) (r=0.333, p<0.001), Near/Far Switching (NFQSCORE) (r=0.360, p<0.001), Multi-target Tracking Speed (MOTSPEED) (r=0.351, p<0.001), and Go/No-Go Score (GNGSCORE) (r=0.362, p<0.001). A weaker positive correlation was noted for Multi-target Tracking Object (MOT_OBJ) (r=0.273, p<0.001).

On the negative side, a strong negative correlation was found between Near/Far Switching Reaction Time (NFQFRT) and performance (r=−0.510, p<0.001), suggesting that slower reaction times are detrimental. Target Capture (TC) also showed a high negative correlation (r=−0.403, p<0.001). Moderate negative correlations were seen for Depth Perception (DPP) (r=−0.377, p<0.001), Eye-Hand Coordination (EHCRT, EHCCRT) (r=−0.425 and r=−0.412, respectively, p<0.001), and Go/No-Go Number of Incorrect Clicks (GNGRHIT) (r=−0.241, p<0.01).

No significant correlations were found for Visual Clarity (VCL, VCB), Contrast Sensitivity (CS6, CS18), Depth Perception (DPL, DPR), Near-Far Proximal Reaction Time (NFQNRT), and Reaction Time (RTA, RTND), as well as several other metrics related to multi-target tracking and hand-eye coordination.

Table 2 shows the association between visual ability indicators and specialized performance. Significant high correlations were found with PDLD (r=−0.54, p<0.01), FDHS (r=0.51, p<0.01), FDHD (r=0.542, p<0.01), and FDLD (r=0.563, p<0.01), explaining 26.0% − 36.7% of the variance in performance. Moderate correlations were observed with FCHS (r=−0.452, p<0.01), FCHD (r=−0.325, p<0.01), FCLD (r=−0.493, p<0.01), ASALS (r=−0.473, p<0.01), ASAHD (r=−0.306, p<0.01), ASALD (r=−0.433, p<0.01), PDHD (r=−0.362, p<0.01), and BCLD (r=−0.442, p<0.01), explaining 10.6% − 24.3% of the variance. Weak correlations were found with RTHS (r=−0.144, p<0.05) and RTLD (r=−0.238, p<0.01), explaining 2.1% − 5.7% of the variance.

**Table 2. Correlation between indicators and performance in the test of visual ability in specialized sports.**

| Predictive Variables | r | Y Specific Achievements | | | | |
|---|---|---|---|---|---|---|
| | | R² | Calibrate R² | Durbin Watson | F | t |
| FD_HS/s | .510** | 0.260 | 0.257 | 1.991 | 73.211*** | 8.556*** |
| FD_LS/s | 0.099 | 0.010 | 0.005 | 2.016 | 2.071 | 1.439 |
| FD_HD/s | .542** | 0.294 | 0.290 | 1.919 | 86.490*** | 9.300*** |
| FD_LD/s | .563** | 0.317 | 0.314 | 1.945 | 96.644*** | 9.831*** |
| FC_HS/time | −.452** | 0.205 | 0.201 | 1.923 | 53.530*** | −7.316*** |
| FC_LS/time | −0.112 | 0.012 | 0.008 | 2.027 | 2.623 | −1.619 |
| FC_HD/time | −.325** | 0.106 | 0.101 | 2.003 | 24.594*** | −4.959*** |
| FC_LD/time | −.493** | 0.243 | 0.240 | 1.864 | 66.845*** | −8.176*** |
| ASA_HS/px | −0.093 | 0.009 | 0.004 | 2.013 | 1.817 | −1.348 |
| ASA_LS/px | −.473** | 0.223 | 0.220 | 1.850 | 59.840*** | −7.736*** |
| ASA_HD/px | −.306** | 0.093 | 0.089 | 1.959 | 21.443*** | −4.631*** |
| ASA_LD/px | −.433** | 0.188 | 0.184 | 1.974 | 48.002*** | −6.928*** |
| PD_HS/mm | −0.022 | 0.000 | −0.004 | 2.029 | 0.098 | −0.314 |
| PD_LS/mm | −0.097 | 0.009 | 0.005 | 2.034 | 1.967 | −1.402 |
| PD_HD/mm | −.362** | 0.131 | 0.127 | 2.115 | 31.402*** | −5.604*** |
| PD_LD/mm | −.540** | 0.291 | 0.288 | 1.992 | 85.401*** | −9.241*** |
| BC_HS/time | −0.078 | 0.006 | 0.001 | 2.022 | 1.289 | −1.135 |
| BC_LS/time | −0.083 | 0.007 | 0.002 | 2.042 | 1.442 | −1.201 |
| BC_HD/time | −0.090 | 0.008 | 0.003 | 2.030 | 1.685 | −1.298 |
| BC_LD/time | −.442** | 0.195 | 0.191 | 1.924 | 50.444*** | −7.102*** |
| RT_HS/ms | −.144* | 0.021 | 0.016 | 2.030 | 4.377 | −2.092 |
| RT_LS/ms | −0.083 | 0.007 | 0.002 | 2.047 | 1.444 | −1.202 |
| RT_HD/ms | −0.065 | 0.000 | −0.005 | 2.029 | 0.006 | −0.080 |
| RT_LD/ms | −.238** | 0.057 | 0.052 | 2.031 | 12.537*** | −3.541*** |

## Performance predictors in clay target Shooting Specialties

In Table 3, specialized performance was treated as the dependent variable. Significant test indicators from a one-way linear regression analysis were used as independent variables in a multiple linear stepwise regression. This model explained 76.7% of the variance in specialized performance (corrected $R^2 = 0.752$, $F = 49.692$, $P < 0.001$). Key variables such as NFQ_F_RT, EHC_RT, EHC_C_RT, RT_D, PS, and GNG_SCORE significantly predict performance in the basic motor vision test, while FC_LD, ASA_LS, ASA_LD, PD_LD, FD_HS, FD_HD, and FD_LD significantly influenced it in the specialized motor vision tests. The high explanatory power of the regression model ($R^2 = 0.767$) underscores that visual abilities collectively play a pivotal role in elite skeet shooting performance, even if individual correlations are moderate

$$\hat{Y} = 205.968 - 0.043 \times RT\_D + 0.025 \times PS - 0.016 \times EHC\_C\_RT - 0.056$$
$$\times NFQ\_F\_RT + 0.517 \times GNG\_SCORE - 0.017 \times EHC\_RT + 5.077$$
$$\times FD\_LD + 3.776 \times FD\_HD + 4.884 \times FD\_HS - 1.198$$
$$\times PD\_LD - 0.778 \times FC\_LD - 0.018 \times ASA\_LS - 0.015 \times ASA\_LD$$

## Discussion

This study explored the relationship between the visual skills of elite skeet shooters and their competition performance. A strong positive association was found between Perceived Range (PS) and performance, suggesting that athletes with a higher perceptual range perform better. Moderate positive correlations were also noted for Visual Clarity (VCR), Near/Far Switching (NFQSCORE), Multi-target Tracking Speed (MOTSPEED), and Go/No-Go Score (GNGSCORE). Conversely, a strong negative correlation was identified between Near/Far Switching Reaction Time (NFQFRT) and performance, indicating that slower reaction times hinder performance. Additionally, Target Capture (TC), Depth Perception (DPP), and Eye-Hand Coordination (EHC_RT) displayed moderate negative correlations. The regression model explained 76.7% of the variance in performance, with key predictors including NFQFRT, EHC_RT, and PS.

Our study assessed athletes' visual systems using metrics for static visual acuity (VC_R, VC_L, VC_B) and depth perception (DP_P, DP_L, DP_R). These metrics align with Elmurr's hardware metrics concept [26], which is vital for evaluating motor visual ability. Software metrics relate to how visual information is processed, influenced by the athlete's

**Table 3. Key influencing factors of skeet shooters' special performance via multiple linear stepwise regression analysis.**

| Predictive Variables | R² | Calibrate R² | F | B | β | t |
|---|---|---|---|---|---|---|
| (Constant) | 0.767 | 0.752 | 49.692*** | 205.968 | | 8.741*** |
| RT__D/ms | – | – | – | −0.043 | −0.135 | −3.448** |
| PS/↑ | – | – | – | 0.025 | 0.122 | 3.07** |
| EHC_C_RT/ms | – | – | – | −0.016 | −0.158 | −4.196** |
| NFQ_F_RT/ms | – | – | – | −0.056 | −0.120 | −2.986** |
| GNG__SCORE | – | – | – | 0.517 | 0.113 | 3.016** |
| EHC_RT/ms | – | – | – | −0.017 | −0.090 | −2.323** |
| FD_LD/s | – | – | – | 5.077 | 0.140 | 3.346** |
| FD_HD/s | – | – | – | 3.776 | 0.116 | 2.763** |
| FD_HS/s | – | – | – | 4.884 | 0.141 | 3.582** |
| PD_LD/mm | – | – | – | −1.198 | −0.145 | −3.502** |
| FC_LD/time | – | – | – | −0.778 | −0.098 | −2.447** |
| ASA_LS/px | – | – | – | −0.018 | −0.116 | −2.94** |
| ASA_LD/px | – | – | – | −0.015 | −0.099 | −2.531** |

experience and strategies. We found that metrics like information processing strategies and NFQ_F_RT (related to near/far switching) significantly impacted specialized performance, with NFQ_F_RT showing a negative correlation with performance. Visual clarity and depth perception, as described by Erickson, are key indicators of an athlete's static visual acuity [27]. Better visual clarity and depth perception enable athletes to identify targets and judge distances effectively [28]. Klemish et al. [29] showed that professional batters had superior visual abilities compared to pitchers, confirming that high-level athletes possess enhanced perceptual skills. Our research indicated that elite skeet shooters exhibited strong visual clarity and depth perception, contributing to their performance by improving target identification and distance judgment. Overall, effective prediction of target movements relies on robust foundational visual abilities, supported by the correlations observed between basic visual metrics and specialized performance [30].

Near-far switching and target capture are key measures of dynamic visual acuity (DVA) in our study, reflecting athletes' eye coordination and movement efficiency. In a basic visual ability test, we assessed NFQ_F_RT (near-far switching) and TC (target capture). Quick eye movements in response to fast-moving targets significantly impact DVA measurements and the performance of skeet shooters [31]. Clay targets are thrown at varying speeds, reaching up to 25 meters per second, requiring athletes to have strong dynamic visual sensitivity across different conditions[32]. Poltavski etc. [14] demonstrated that dynamic visual acuity scores can predict goal scoring percentages in ice hockey players. Our findings indicate that DVA-related measures influenced the specialized performance of skeet shooters. Previous studies show that elite athletes generally have better DVA than non-elites [33]. Our research confirmed that elite skeet shooters had superior visual abilities related to DVA compared to non-elite shooters. Additionally, it was suggested that dynamic visual attention training can enhance overall athletic performance [34], highlighting the potential for targeted training in improving the performance of skeet shooters in future studies.

Our study assessed the perceptual range of clay target sports athletes, finding a significant correlation between perceptual range (PS) and specialized performance in skeet shooters. We also evaluated multi-target tracking abilities, discovering that metrics like MOT_SPEED correlated with performance, supporting the assessment's relevance. While the correlation coefficients reported here may appear modest, they align with typical effect sizes in sports science, where performance outcomes are influenced by numerous interacting factors. Even moderate correlations can reflect practically significant relationships in elite athlete cohorts, as small improvements in visual skills may translate to critical competitive advantages. Proficient athletes can pre-store target information in their spatial working memory, enhancing their ability to locate and intercept targets [34]. Our findings showed that elite skeet shooters exhibited strong visual search patterns and quick, accurate information processing linked to their performance.

Reichow's test study (N = 20) found a strong correlation between a hitter's ability to recognize fast pitches and their batting average from the previous season [35]. While our research focused on skeet shooting, this suggests a link between visual recognition skills and performance in various sports. We discovered that the ability to recognize visual information such as through search patterns and acuity—was associated with the performance of skeet shooters. Elite shooters can track targets in real-time, with a wide field of vision and efficient eye movements. Our measurements of gaze duration, frequency, and jump amplitude correlated significantly with performance, emphasizing their importance. Extensive training allows skeet shooters to allocate attention effectively, focusing on crucial visual information. Athletes with better training in these areas tended to perform better [36]. Additionally, the demanding nature of skeet shooting can lead to fatigue [37], which we controlled for in our study. Our findings indicate that maintaining visual skills, even under fatigue, is essential for success, demonstrating a connection between visual abilities and specialized sports achievements.

Reaction time is a key measure of an individual's movement skills efficiency [38]. Our tests indicated that dominant hand reaction time (RT_D) and other metrics had significant correlations with specialized performance, with RT_D showing a negative correlation. Hand-eye coordination combines visual processing and motor execution, while decision-making assessments involve additional judgment in response to visual stimuli [39]. We found that the GNG_SCORE metric for decision-making positively correlated with specialized performance, suggesting that decision-making abilities impact performance. Elite skeet athletes focus on rapid execution, requiring swift reactions in about 0.4 to 0.7 seconds as clay

targets travel at high speed. The visual information and athlete reaction time are crucial for success [40]. Athletes' exceptional proficiency often results from years of structured practice [41], enhancing their hand-eye coordination and decision-making [42]. Our study shows a strong link between visual abilities, performance, and decision-making, highlighting how superior decision-making differentiates elite from sub-elite skeet shooters.

The study's findings indicate that various visual abilitiessuch as visual clarity (VC), contrast sensitivity (CS), depth perception (DP), and decision-making processes (GNG)—impact performance in clay target shooting to varying degrees. For example, near-far switching (NFQ_F_RT) correlated negatively with performance, while visual clarity (VC_R) showed a positive correlation. A central idea in sports vision science is that enhanced visual skills improve athletic performance. Kirschen's visual pyramid [43] and Welford's processing model [44] suggest that success in high-level motor tasks depends on the effective functioning of foundational processes. The study highlights that software aspects of the visual system correlate more strongly with specialized performance in skeet shooting than basic perceptual abilities. This aligns with a study which indicates that software capabilities are better predictors of sports performance [45]. Thus, the ability of top skeet shooters to respond to visual information is primarily linked to their decision-making and execution skills.

Zhang Wenjie, a clay target shooting instructor, has gathered statistics on eight top male and female players across four matches and 26 technical exercises [46]. In 2,500 clay targets, there were 193 misses, including 119 double-target misses, resulting in a miss rate of 61.7% (193 out of 312) and double-target misses at 30.3% (119 out of 312). For members of the Chinese National Shooting Team, 11,125 clay targets yielded 750 misses, with 460 double-target misses, 61.3%. The results of two-way competitions with a total of 11125 targets were similar, with 750 misses and 460 double-targets misses at the same rate. High-level athletes distinguish themselves with rapid launching speeds. The swift and fluid style of "fast-playing" shooters is a trend in skeet shooting technology [47]. This fast-hitting technique allows athletes to minimize distractions and improve accuracy [48]. It requires a 'see to drive' approach, where visual perception guides their actions, emphasizing the importance of visual search abilities in dual-target scenarios.

The indicators of gaze frequency and duration reflect athletes' visual search processes. Gaze frequency indicates proficiency and strategy; lower frequency suggests greater efficiency, while longer duration implies deeper information processing. Our study found a significant correlation between gaze indicators and specialized performance among skeet shooters. We measured gaze duration across different scenarios, including high-table and low-table single and dual targets (FD_HS, FD_LS, FD_HD, FD_LD). Each scenario showed distinct correlations with performance, highlighting the importance of gaze duration. Abernethy's research suggests that elite athletes use a "low search rate" to minimize cognitive load, allowing for effective information extraction with fewer gazes [49]. High-level clay target shooters must quickly detect and maintain focus on targets, resulting in characteristics like low-frequency concentration and stable gaze patterns [50]. Williams' study found that expert football players exhibited less frequent gazes [51], while Helsen and Pauwels noted that experienced athletes utilized longer gaze durations and focused on relevant information [52]. Xiong Jianping's research on volleyball players showed that experts had significantly longer gaze durations during blocking scenarios [53], indicating superior information processing abilities compared to less experienced athletes.

During the visual search process, athletes must quickly locate relevant information by shifting their gaze, a movement known as "eye hopping" [54]. Our study shows that the amplitude of these eye jumps significantly correlates with specialized performance. We measured the average eye jump (ASA) across various scenarios: high-table single-target (ASA_HS), low-table single-target (ASA_LS), high-table dual-target (ASA_HD), and low-table dual-target (ASA_LD). The results indicate that different eye jump amplitudes affect performance among skeet shooters. Notably, a smaller eye jump distance is linked to higher performance, reflecting the straightforward nature of clay target sports. Due to their superior information integration abilities, expert athletes can efficiently extract critical information with smaller eye jumps. When combined with longer gaze durations, these athletes can locate key information more quickly and at a deeper level, enhancing their decision-making agility. Jiancheng Zhang's research on table tennis players found that elite athletes used a lower frequency of gaze and shorter eye-hopping distances when observing rotational serves, a phenomenon known as "quiet eye," [54,55] which helps

maintain focus on the source of information. Our findings support the idea that favorable eye jump amplitudes correlate with better performance, consistent with established visual search strategies in elite athletes [56].

Blinking and pupil size provide insights into cognitive engagement. When the brain needs more resources, pupils dilate [57]. In high-intensity tasks like skeet shooting, fewer blinks enhance reaction time and accuracy by maintaining continuous visual input and preventing visual blindness at critical moments. Athletes develop specialized visual processing strategies, evident in their brain activity [58]. For example, a study by Zhang et al. found that expert volleyball players had smaller pupil diameters and less prefrontal cortex activation than novices, indicating efficient visual processing with less cognitive load [59]. Similarly, elite clay target shooters show reduced prefrontal cortex activity and smaller pupil diameter changes [60], reflecting their efficient information processing and quicker predictions [61,62]. The empirical dominance hypothesis suggests that expert athletes' advantages come from developed pattern recognition and anticipation skills, leading to superior performance [63]. Comparing our findings on blinking and pupil size with existing literature shows that these physiological responses significantly impact performance in visual tasks. This highlights the potential for future research to optimize training programs for athletes based on these insights.

The integration of Senaptec (basic vision) and aSee Glasses (specialized vision) highlights that elite performance depends not only on foundational visual skills but also on their application in dynamic, task-specific contexts. Furthermore, these data analyses employed in this study can be used in the future to develop visual training programs for targeting skeet shooters, helping to improve their visual stability and tracking ability and to create more cost-effective and efficient visual search patterns, thus contributing to improvements in specialized performance. The study was carried out in a particular laboratory environment. Additionally, the sample size of 42 skeet shooters, although is not adequate for some analyses, may not be sufficient to capture all possible variations in visual abilities and performance. A larger sample could potentially yield more comprehensive and accurate results. The ecological validity concern implies that the relationships between visual abilities and identified performance may not be identical in competition settings. The limited sample size might result in some undetected relationships or inaccuracies in the key factors affecting performance. Future research should focus on conducting studies in more naturalistic settings to improve ecological validity and increase the sample size for more reliable and generalizable results.

## Conclusion

The specialized performance of elite skeet shooters is influenced by both basic motor-visual abilities and specialized visual search patterns. Specifically, the capacity to process visual information within the basic motor-visual abilities is more crucial for enhancing specialized performance than the mere reception of visual stimuli. Furthermore, the visual search pattern involving two targets within the specialized visual search pattern has a more significant role in enhancing the specialized performance of skeet shooters. Future research could focus on further exploring the mechanisms underlying these relationships and developing more targeted training programs based on these findings.

## Supporting information

**S1 Appendix. Description of Motor Visual Test Indicators of Chinese National Skeet Shooters.**
(DOCX)

**S2 Table. Specialized Scores and the Indicators of the Basic. Motor Visual Ability Test.**
(DOCX)

**S3 Table. Indicators of the Specialized Visual Ability Test.**
(DOCX)

**S4 File. Raw data.**
(XLSX)

## Acknowledgments

The authors would like to thank Xiao Wang for their technical support throughout the project. The authors would also like to thank all the shooters who participated in this study.

## Author contributions

**Conceptualization:** Dongxu Gao, Yang Wu, Chao Chen.

**Data curation:** Dongxu Gao, Beishi Hu, Tinggang Yuan.

**Formal analysis:** Dongxu Gao, Qingshou Guo, Pengfei Wei, Yang Wu, Chao Chen.

**Funding acquisition:** Dongxu Gao.

**Investigation:** Dongxu Gao, Beishi Hu, Tinggang Yuan, Yang Wu.

**Methodology:** Dongxu Gao, Beishi Hu, Tinggang Yuan, Yang Wu, Chao Chen.

**Project administration:** Dongxu Gao, Chao Chen.

**Resources:** Dongxu Gao, Chao Chen.

**Software:** Dongxu Gao, Beishi Hu.

**Supervision:** Dongxu Gao, Chao Chen.

**Validation:** Dongxu Gao.

**Visualization:** Dongxu Gao, Tinggang Yuan.

**Writing – original draft:** Dongxu Gao, Beishi Hu, Tinggang Yuan, Yang Wu.

**Writing – review & editing:** Dongxu Gao, Qingshou Guo, Pengfei Wei, Chao Chen.

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
