## [Editor Report · Decision Letter 0]

28 Jan 2025

PONE-D-24-54464Exploring the Relationship between Motor Visual Proficiency and Performance Metrics in Elite Skeet Shooters: An In-depth AnalysisPLOS ONE

Dear Dr. Wu,

Thank you for submitting your manuscript to PLOS ONE. After careful consideration, we feel that it has merit but does not fully meet PLOS ONE’s publication criteria as it currently stands. Therefore, we invite you to submit a revised version of the manuscript that addresses the points raised during the review process.

We look forward to receiving your revised manuscript.

Kind regards,

Nick Fogt

Academic Editor

PLOS ONE

Journal Requirements:

“Funding

Dongxu Gao and Tinggangyuan

China Institute of Sport Science Basic Scientific Research Business Funding Project,(Basic 23-39)”

“The authors have stated that there is no competing interest.”

6. We note that Figures 3 and 5 includes an image of a participant in the study. 

Additional Editor Comments:

My apologies for the delay in looking for reviewers for this paper.

Prior to sending for review, I am requesting that the authors please fix the formatting of the references. The formatting of the references is highly variable, and there is at least one duplicate.

As soon as this is corrected, I will begin the process of finding reviewers.

---

## [Author Response · Author response to Decision Letter 1]

12 Feb 2025

PONE-D-24-54464R1: PLOS ONER1

Title: Exploring the Relationship between Motor Visual Proficiency and Performance Metrics in Elite Skeet Shooters: An In-depth Analysis

Article Type: Research Article

Dear Editor

Response to the review comments:

Thank you for the useful comments and suggestions on our manuscript. We have included required information and revised the manuscript accordingly, and detailed corrections are listed below:

Response to comments by Academic Editor:

We have rechecked and corrected the mistakes as suggestion given.

2. Thank you for stating the following financial disclosure: “Funding Dongxu Gao and Tinggangyuan China Institute of Sport Science Basic Scientific Research Business Funding Project,(Basic 23-39)” Please state what role the funders took in the study. If the funders had no role, please state: "The funders had no role in study design, data collection and analysis, decision to publish, or preparation of the manuscript.

"We have added the funding information as suggestion given”

{Please refer to highlights in the cover letter]

“The authors have stated that there is no competing interest.” Please complete your Competing Interests on the online submission form to state any Competing Interests. If you have no competing interests, please state "The authors have declared that no competing interests exist.", as detailed online in our guide for authors at http://journals.plos.org/plosone/s/submit-now. This information should be included in your cover letter; we will change the online submission form on your behalf.

"We have added the competing interest information as suggestion given”

{Please refer to highlights in the cover letter]

4. PLOS requires an ORCID iD for the corresponding author in Editorial Manager on papers submitted after December 6th, 2016. Please ensure that you have an ORCID iD and that it is validated in Editorial Manager.

“The corresponding author has added ORCID in the submission account as suggestion given.”

5. Please include captions for your Supporting Information files at the end of your manuscript, and update any in-text citations to match accordingly.

We have included captions for supporting information files at the end of the manuscript as suggested.

[Please refer to highlights at the end of the manuscript]

6. We note that Figures 3 and 5 includes an image of participants.

We have covered the identifiable parts of the study participants in figure 3 and 5.

[Please refer to figure 3 and figure 5]

Sincerely

Yang Wu

---

## [Editor Report · Decision Letter 1]

14 Feb 2025

PONE-D-24-54464R1Exploring the Relationship between Motor Visual Proficiency and Performance Metrics in Elite Skeet Shooters: An In-depth AnalysisPLOS ONE

Dear Dr. Wu,

Thank you for submitting your manuscript to PLOS ONE. After careful consideration, we feel that it has merit but does not fully meet PLOS ONE’s publication criteria as it currently stands. Therefore, we invite you to submit a revised version of the manuscript that addresses the points raised during the review process.

We look forward to receiving your revised manuscript.

Kind regards,

Nick Fogt

Academic Editor

PLOS ONE

**Additional Editor Comments:**

The references still are not uniform. Please follow the instructions to authors in uniformly formatting all of the references. I won't be able to look for reviewers until this matter is resolved.

---

## [Author Response · Author response to Decision Letter 2]

15 Feb 2025

PONE-D-24-54464R2: PLOS ONER2

Title: Exploring the Relationship between Motor Visual Proficiency and Performance Metrics in Elite Skeet Shooters: An In-depth Analysis

Short title: Motor Visual Skills and Performance in Elite Skeet Shooters

Article Type: Research Article

Dear Editor

Response to the review comments:

Thank you for the useful comments and suggestions on our manuscript. We have included required information and revised the manuscript accordingly, and detailed corrections are listed below:

Response to comments by Academic Editor:

1. The references still are not uniform. Please follow the instructions to authors in uniformly formatting all of the references. I won't be able to look for reviewers until this matter is resolved.

Dear editor, thank you for your feedback on references. We have rechecked and amended the entire references and citations based on the instructions to authors as suggested.

[Please refer to highlights in the reference section]

2. Please upload your figure files to the Preflight Analysis and Conversion Engine (PACE) digital diagnostic tool.

We have uploaded all figures to PACE tool, all figures were improved accordingly.

Sincerely

Yang Wu

---

## [Decision Letter · Decision Letter 2]

27 Apr 2025

PONE-D-24-54464R2Exploring the Relationship between Motor Visual Proficiency and Performance Metrics in Elite Skeet Shooters: An In-depth AnalysisPLOS ONE

Dear Dr. Wu,

Thank you for submitting your manuscript to PLOS ONE. After careful consideration, we feel that it has merit but does not fully meet PLOS ONE’s publication criteria as it currently stands. Therefore, we invite you to submit a revised version of the manuscript that addresses the points raised during the review process.

We look forward to receiving your revised manuscript.

Kind regards,

Nick Fogt

Academic Editor

PLOS ONE

Journal Requirements:

Additional Editor Comments:

Both reviewers have positive things to say about the paper.

Please comment on reviewer #1's comments regarding the statistical analysis. While the overall r squared value is quite good, the correlation coefficients between various tests are somewhat low. Please comment on the implications of these lower correlations. Reviewer #1 also asks for some details around the aSee glasses. Presumably these glasses are independent from the Senaptec system. If the aSee glasses are not related to the Senaptec system please indicate that this is the case. Please provide the manufacturer of the aSee system, the resolution (which looks like 0.50deg from the company website (https://www.7invensun.com/elite)) and if known, the accuracy of this eye tracking system.

Reviewers' comments:

Reviewer's Responses to Questions

**Comments to the Author**

1. If the authors have adequately addressed your comments raised in a previous round of review and you feel that this manuscript is now acceptable for publication, you may indicate that here to bypass the “Comments to the Author” section, enter your conflict of interest statement in the “Confidential to Editor” section, and submit your "Accept" recommendation.

Reviewer #1: All comments have been addressed

Reviewer #2: All comments have been addressed

2. Is the manuscript technically sound, and do the data support the conclusions?

Reviewer #1: Yes

Reviewer #2: Yes

3. Has the statistical analysis been performed appropriately and rigorously? 

Reviewer #1: N/A

Reviewer #2: Yes

4. Have the authors made all data underlying the findings in their manuscript fully available?

Reviewer #1: Yes

Reviewer #2: Yes

5. Is the manuscript presented in an intelligible fashion and written in standard English?

Reviewer #1: Yes

Reviewer #2: Yes

6. Review Comments to the Author

Reviewer #1: I think that the paper is good but I don´t understand statistical about r because is relatively low. I like senaptec system but I don't see the relation with Asee glasses, Can you explain a little more?

Reviewer #2: The text presents a detailed analysis of the importance of visual capabilities in shooting performance. It is clear that faster occurrence times and better cooperative insight are crucial for superior performance. The negative correlation between occurrence times and performance highlights that slower occurrence times are relevant to athletes, emphasizing the need to focus on developing these skills. Furthermore, the regression model that explains 76.7% of the variance in expert performance suggests that visual predictors are substantial for success in sport. The conclusion reaffirms the importance of perceived breadth and specialized skills, ensuring that training in these aspects can bring significant improvements in competitors' performance.

So congratulations to the authors.

7. PLOS authors have the option to publish the peer review history of their article (what does this mean? ). If published, this will include your full peer review and any attached files.

**Do you want your identity to be public for this peer review?** For information about this choice, including consent withdrawal, please see our Privacy Policy .

Reviewer #1: **Yes: ** Ricardo Bernárdez-Vilaboa

Reviewer #2: No

---

## [Author Response · Author response to Decision Letter 3]

3 May 2025

PONE-D-24-54464R2: PLOS ONER3

Title: Exploring the Relationship between Motor Visual Proficiency and Performance Metrics in Elite Skeet Shooters: An In-depth Analysis

Short title: Motor Visual Skills and Performance in Elite Skeet Shooters

Article Type: Research Article

Dear Editor

Response to the review comments

Thank you for the useful comments and suggestions on our manuscript. We have included the required information and revised the manuscript accordingly, and a point-by-point response to review questions is listed below:

Response to journal requirements

1. Please review your reference list to ensure that it is complete and correct. If you have cited papers that have been retracted, please include the rationale in the manuscript text or remove these references and replace them with relevant current references. The rebuttal letter should mention any changes to the reference list that accompany your revised manuscript. If you need to cite a retracted article, indicate the article's retracted status in the References list and include a citation and full reference for the retraction notice.

Response: We have checked all references for retraction and completeness, and no retracted articles were identified on the list. All references are currently valid as per Retraction Watch and PubMed checks. We have also replaced some blogs and duplicate references with recent empirical studies.

Response to comments by Academic Editor

1. Please comment on reviewer #1's comments regarding the statistical analysis. While the overall r-squared value is quite good, the correlation coefficients between various tests are somewhat low. Please comment on the implications of these lower correlations. Reviewer #1 also asks for some details about the aSee glasses. Presumably, these glasses are independent of the Senaptec system. If the aSee glasses are not related to the Senaptec system, please indicate this.

Response: Dear editor, Thank you for your overall feedback. We have addressed reviewer 1's comments. The detailed point-by-point response is mentioned under response to reviewer 1 comments below and highlighted in the method, results, and discussion section of the manuscript.

2. Please provide the manufacturer of the aSee system, the resolution (which looks like 0.50deg from the company website (https://www.7invensun.com/elite)), and, if known, the accuracy of this eye tracking system.

Response: Thank you for requesting specification details. We have addressed this comment and provided the details regarding the aSee eye-tracking system, as suggested.

Please refer to highlights in the method section of the manuscript where we have addressed this comment.

[…The specialized visual ability test utilized the aSee Glasses (7INVENSUN, Beijing, China), a spectacle-type eye-tracking device with a resolution of 0.50° and an accuracy of 0.5°–1.0°. The system’s high-frequency sampling rate (120 Hz) and precision make it suitable for capturing dynamic gaze behaviors in sport-specific scenarios….]

Response to comments by Reviewer #1

1. I don't understand statistics about r because it is relatively low.

Response: Thank you for this insightful clarification question and comments. While the correlation coefficients (r-values) reported in our study (e.g., 𝑟 =0.486 for Perceived Range) may appear modest, they are consistent with findings in sports science and human performance research, where multifactorial influences (e.g., psychological, physiological, environmental) often result a significant effect on performanes. Even moderate correlations (e.g., r=0.3–0.5) are meaningful in such contexts, reflecting measurable and actionable relationships between variables. For example, r=0.486 (Perceived Range vs. performance) explains ~23.6% of the variance (𝑅2 =0.237), which is substantial in elite athlete populations where marginal gains are critical.

[Please refer to highlights in the discussion part where we have addressed this comment]

2. I like the Senaptec system but do not see the relation with aSee Glasses. Can you explain a little more?

Response: The Senaptec system assesses basic motor visual abilities (e.g., visual clarity, depth perception, reaction time) under controlled conditions, providing foundational metrics of athletes’ visual hardware. In contrast, the aSee Glasses evaluate specialized visual abilities (e.g., gaze patterns, target tracking, decision-making) during dynamic, sport-specific tasks, reflecting how athletes apply these foundational skills in ecologically valid scenarios. Together, they provide a comprehensive evaluation: Senaptec identifies baseline visual capabilities, while aSee Glasses reveals how these capabilities translate to performance under competition-like demands.

[Please refer to the highlights in the materials and method section where we explained this comment.]

Sincerely

Yang Wu

---

## [Editor Report · Decision Letter 3]

13 May 2025

Exploring the Relationship between Motor Visual Proficiency and Performance Metrics in Elite Skeet Shooters: An In-depth Analysis

PONE-D-24-54464R3

Dear Dr. Wu,

We’re pleased to inform you that your manuscript has been judged scientifically suitable for publication and will be formally accepted for publication once it meets all outstanding technical requirements.

Kind regards,

Nick Fogt

Academic Editor

PLOS ONE

Additional Editor Comments (optional):

Thank you for your responses to the reviewer comments.
---

## [Editor Report · Acceptance letter]

PONE-D-24-54464R3

PLOS ONE

Dear Dr. Wu,

I'm pleased to inform you that your manuscript has been deemed suitable for publication in PLOS ONE. Congratulations! Your manuscript is now being handed over to our production team.

Kind regards,

on behalf of

Dr. Nick Fogt

Academic Editor

PLOS ONE